# Size scaling of large landslides from incomplete inventories

Oliver Korup [1,2], Lisa V. Luna [1,3], and Joaquin V. Ferrer [1,4]

[1]Institute of Environmental Sciences and Geography, University of Potsdam, Germany
[2]Institute of Geosciences, University of Potsdam, Germany
[3]U.S. Geological Survey, Golden CO, United States
[4]Potsdam Institute for Climate Impact Research PIK, Germany

**Correspondence:** Oliver Korup  (korup@uni-potsdam.de)

**Abstract.** Landslide inventories have become cornerstones for estimating the relationship between the frequency and size of slope failures, thus informing appraisals of hillslope stability, erosion, and commensurate hazard. Numerous studies have reported how larger landslides are systematically rarer than smaller ones, drawing on probability distributions fitted to mapped landslide areas or volumes. In these models, much uncertainty concerns the larger landslides (defined here as affecting areas $\geq 0.1$ km$^2$) that are rarely sampled, and often projected by extrapolating beyond the observed size range in a given study area. Relying instead on size-scaling estimates from other inventories is problematic because landslide detection and mapping, data quality, resolution, sample size, model choice, and fitting method can vary. To overcome these constraints, we use a Bayesian multi-level model with a Generalised Pareto likelihood to provide a single, objective, and consistent comparison grounded in extreme-value theory. We explore whether and how scaling parameters vary between 37 inventories that, although incomplete, bring together 8627 large landslides. Despite the broad range of mapping protocols and lengths of record, and differing topographic, geological, and climatic settings, the posterior power-law exponents remain indistinguishable between most inventories. Likewise, the size statistics fail to separate known earthquake from rainfall triggers, and event-based from multi-temporal catalogues. Instead, our model identifies several inventories with outlier scaling statistics that reflect intentional censoring during mapping. Our results thus caution against a universal or solely mechanistic interpretation of the scaling parameters, at least in the context of large landslides.

## 1 Introduction

Keeping records of the size and frequency of landslides is key to estimate rates of erosion, geomorphic work, and hillslope evolution (Dente et al., 2023; Saito et al., 2014; Marc et al., 2019); infer material strength and weathering of hillslopes (Li and Moon, 2021; Alberti et al., 2022), and inform hazard appraisals of slope instability (Guzzetti et al., 2012), particularly in response to contemporary climate change (Smith et al., 2023). A popular way to characterise the relative frequency of landslide-affected areas or volumes is to fit probability distributions to size data compiled in inventories (Malamud et al., 2004). These

catalogues contain locations and geometries of individual footprint areas mapped largely from air photos or satellite images. The choice of probability distribution (or "scaling laws") has favoured the inverse power-law or Pareto, the inverse gamma, or the lognormal distributions (Tebbens, 2020), or combinations thereof (Jain et al., 2022). All these distributions are skewed, often heavy-tailed, and capture the widespread observation that larger landslides are systematically rarer than smaller ones.

Reported values of the parameters that define these distributions have seemingly narrow numerical ranges (Tebbens, 2020). This similarity among model fits has led to a lively discussion about whether these parameters reflect generic geometric or mechanistic properties of landslides or the hillslopes that they occur on (Bellugi et al., 2021; Bernard et al., 2021). For example, physical interpretations of the "roll-over" that marks the lower bound of inverse power-law distributions include that of a hillslope length scale that is susceptible to failure, or the cohesive strength of failure planes (Tebbens, 2020). While landslide size distributions may reflect the nature and spatial intensity of a common trigger such as a strong earthquake (Valagussa et al., 2019), the average landslide size may contain information about both cohesive strength of slope material and hillslope relief (Medwedeff et al., 2020). Some of these physical interpretations are backed by, or derived from, numerical simulations of slope instability (Frattini and Crosta, 2013). Yet, a different line of arguments proposed that the "roll-over" is a statistical artefact of landslide detection and mapping, approximately marking the smallest discernible landslide in a given study area (Tebbens, 2020). Either way, this discussion has questioned whether these scaling laws are universally applicable to landslides irrespective of environmental setting, mapping methods, and trigger mechanisms (Malamud et al., 2004; Tanyaş et al., 2019). Many of these interpretations have relied on the direct comparison of reported parameter values, and scrutiny concerning possible effects of data sources and quality, mapping method, and statistical errors of the fitted models has increased in more recent work (Bellugi et al., 2021).

Still, most uncertainty remains about the large landslides that are rarely sampled. Hence, the bulk of studies on landslide size has disclosed little about these large landslides, let alone their prediction as first-time failures (Fan et al., 2019). One reason for this knowledge gap is that large landslides are often elusive in catalogues compiled shortly after a landslide-triggering earthquake or rainstorm (Hao et al., 2020; Abancó et al., 2021; Santangelo et al., 2023). Sample sizes often involve only a handful to several dozen large landslides, and thus often remain too small for robust statistics in a given study area. Hence, inference is mostly based on the simple extrapolation of model fits beyond the observed size range. Yet, large landslides may often re-shape hillslope geometry and dominate erosion (Korup et al., 2007; Marc et al., 2019), but may involve phases of creep motion, and respond differently to triggering conditions than smaller failures because of a longer and more complex slope history of accumulated stress and strain (Lacroix et al., 2020).

In general, the statistics of landslides derive from a sequence of detection, mapping, and statistical inference (Figure 1). Uncertainties that propagate throughout each step can affect the outcome in terms of landslide scaling statistics.

At the level of the input data, both landslide detection and mapping face several constraints. The mapping objective can dictate whether to focus on landslides attributed to a single trigger such as a strong earthquake (Meunier et al., 2013; Gorum et al., 2014; Tanyaş et al., 2017), a rainstorm (Hao et al., 2020; Emberson et al., 2022; Santangelo et al., 2023), or instead to compile landslide traces that have accumulated over years to millennia (LaHusen et al., 2016; Luetzenburg et al., 2022; Fusco et al., 2023). Some of the most comprehensive catalogues today feature thousands to hundreds of thousands of slope

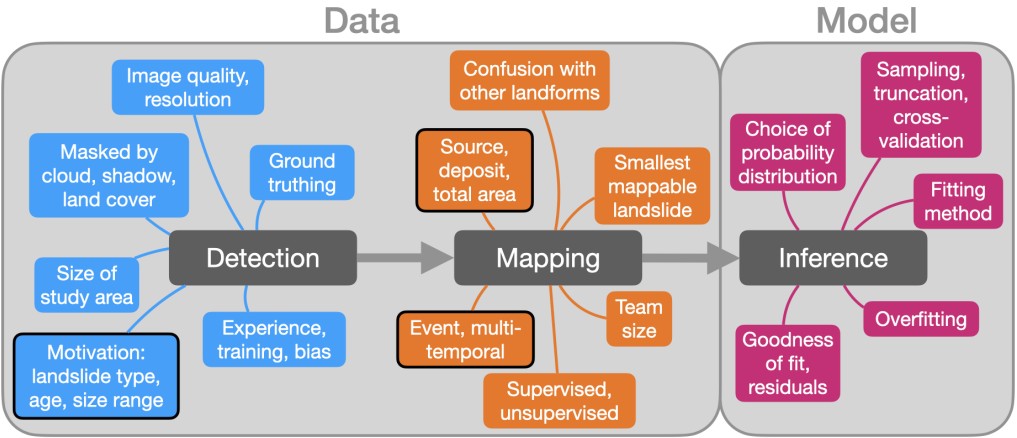

**Figure 1.** Landslide scaling statistics rely on accurate detection, mapping, and statistical inference. These three (colour-coded) steps are prone to a number of uncertainties at the level of both data (i.e. landslide areas from inventories covering specific single trigger events or integrating observations over longer time spans) and model; these uncertainties propagate as errors into the derived landslide statistics. Boxes with black outline are most likely directly tied to physical processes of landsliding.

failures across entire nations or beyond (Luetzenburg et al., 2022; Fusco et al., 2023). The methods to detect, map, and compile landslide inventories have become more diverse and elaborate beyond traditional mapping from air photos, optical satellite data or historical records (Xu et al., 2020; Casagli et al., 2023). Newer catalogues derive from laser scanning (Bernard et al., 2021); radar imagery (Song et al., 2022); object-based image analysis (Milledge et al., 2022); deep neural networks (Schönfeldt et al., 2022); text mining (Franceschini et al., 2022); and seismology (Hibert et al., 2019). Most methods require specific mapping protocols adjusted to the effective resolution of imagery that may be compromised by vegetation, land cover, cloud, and shadow (Brardinoni et al., 2003; Burrows et al., 2022). Debris- flows and snow-avalanche tracks, moraines, and wind-throw gaps in forests can be mistaken for landslide evidence. Overlapping landslide source areas or bodies can obscure the dimensions of slope failure (Marc and Hovius, 2015), and so can subsequent erosion or deposition. Varying image quality, resolution, and coverage all affect landslide size estimates, as does the experience of mapping operators (Van Den Eeckhaut et al., 2005). The mapping outcome may depend on whether a single person or a team is at work, and on the attention to detail in delineating source, deposit, or total affected areas. Experience and training aid detection and mapping, but also introduce bias, for example favouring fresh landslides, certain types of failure, or simply those that are easiest to recognise in case of limited time or training. Hence, the size range of a given inventory is bounded by the smallest mappable, and the largest recognisable, landslide (Barlow et al., 2012). Inventories can be incomplete in that they miss out on those large landslides that have indistinct or less obvious geomorphic evidence, and thus require experience, skill, and time for accurate detection and mapping. Thus,

without any standard mapping protocol in place, landslide researchers have to deal with catalogues of varying extent, detail, and quality for the same task of obtaining traits of landslide size.

At the model level, estimates of size scaling hinge on the choice of probability distribution to characterise the mapped landslides (ten Brink et al., 2009). These estimates also depend on sample size, data pre-processing, fitting method, residuals, and cross-validation. Numerical experiments show that small sample sizes yield volatile estimates of scaling parameters for inverse power-law distributions in particular (Korup et al., 2012). Most estimation methods involve either regression of log-binned - and thus smoothed - landslide frequencies versus size (Gilham et al., 2018), or maximum likelihood estimates (Clauset et al., 2009); various biases apply to both methods. However, reports of confidence intervals or goodness of fit, and hence ways to assess overfitting remain rare. Still, the basis for statistical inference of landslide size distributions varies at the level of the individual inventory, each of which embodies the methods of detection and mapping used, and the biases they may induce. In the light of these constraints, a direct comparison of landslide scaling estimates between different inventories may be misleading.

We propose a compact solution to compare more fairly the landslide-size distributions from diverse inventories by estimating scaling parameters with a single, probabilistically consistent model. We apply this model to large landslides that affect a total area of >0.1 km$^2$, and address the problem of small sample size by using Bayesian inference in a multi-level model that uses data from multiple inventories to estimate the variance of scaling parameters within and across these catalogues (Luna and Korup, 2022). The multi-level approach acknowledges structure in landslide size data in a consistent way. One intuitive grouping of data is by inventory, and reflects the diversity in data input quality reviewed above. Our focus on large landslides makes the Generalised Pareto Distribution (GPD) a natural model choice, because extreme-value theory predicts that data above a high threshold are approximately GP distributed (Castro-Camilo et al., 2022). Another advantage of this distribution is that its parameters can be translated directly into those used most widely in studies of landslide size scaling.

## 2 Data and Methods

We consider data on total landslide-affected areas from several dozen published landslide inventories with open access. We excluded many other detailed catalogues that had no records of landslides meeting our size threshold of 0.1 km$^2$. Besides information about their size, many databases have landslide types and triggers reported, and many data were recorded following recent (i.e. post-1900) major earthquakes and rainstorms with the intention to characterise the impact of these events. We also included catalogues spanning time intervals of several years to millennia, featuring mostly undated large landslides with unknown triggers to test whether these cumulative inventories have size distributions that differ from those of event-based inventories.

The choice of probability distribution to model landslide area often rests on implicit assumptions. For example, the inverse power law draws on considerations of physical sand-pile models and the concept of self-organised criticality (Hergarten and Neugebauer, 1998), whereas the lognormal distribution arises naturally from multiplicative effects of random variables (ten Brink et al., 2009). Here we model reported areas of large landslides with the Generalized Pareto Distribution (GPD). The GPD

is rooted in extreme-value theory and approximates the distribution of a continuous random variable $x$ above a high threshold (or location parameter) $\mu$. The GPD thus captures what we would expect theoretically from a sample consisting of observations filtered above a minimum value (Katz et al., 2002). Any physical interpretation of the GPD parameters may need to account, or correct, for this statistical expectation first. The probability density function of the GPD is:

$$
\text{GPD}(x|\mu, \sigma, k) = \frac{1}{\sigma} \left( 1 + \frac{k(x - \mu)}{\sigma} \right)^{-1/k - 1}, \tag{1}
$$

where $x \geq \mu$, $\sigma > 0$ is a scale parameter, and $k \geq 0$ is a shape parameter. The scale parameter $\sigma$ is somewhat comparable to the "roll-over" in studies using an inverse power-law model for estimating landslide size scaling. This "roll-over" approximates the modal landslide size, which is the smallest landslide size above which power-law scaling is assumed. The GPD shape parameter is the inverse of the "scaling exponent" $\alpha$ of the inverse power-law distribution, such that $k = 1/\alpha$.

Here, the location parameter $\mu$ sets the minimum landslide size for data to be admitted to the GPD, and known as a peak-over-threshold approach in extreme-value statistics (Katz et al., 2002). For large landslides, we let $\mu = 0.1$ km$^2$. Empirical relationships between landslide volume and total affected area across a wide range of environmental settings show that an area of 0.1 km$^2$ corresponds to an average volume of roughly $10^6$ m$^3$ (Larsen et al., 2010), which is the suggested lower threshold for large landslides (McColl and Cook, 2024). This particular choice of $\mu$ is a compromise, because fewer samples and landslide inventories are available for higher values of $\mu$, whereas the GPD becomes a less and less valid approximation to the data for lower values of $\mu$.

Fitting the GPD to data can involve maximum likelihood estimates or, in case of few samples ($n < 30$), probability-weighted (L-)moments to avoid volatile parameter estimates (Katz et al., 2002). We use Bayesian inference to learn the GPD parameters from the data, acknowledging explicitly that these come from different inventories that reflect different environmental conditions across study areas, and landslides likely detected at varying resolution and mapped with different techniques. A Bayesian treatment of this fitting problem seeks a compromise between a likelihood function and a probability distribution of prior knowledge about the model parameters. This approach obviates the need for binned landslide-size data to use frequency density (Malamud et al., 2004). Instead, we work with the joint probability that is the numerator of Bayes' Rule:

$$
p(\theta|\mathcal{D}) = \frac{p(\mathcal{D}|\theta)p(\theta)}{p(\mathcal{D})}, \tag{2}
$$

where $\theta$ is the vector of model parameters that we wish to update from both the landslide-size data $\mathcal{D}$ and prior knowledge. We use the GPD as the likelihood function $p(\mathcal{D}|\theta)$ and choose (hyper-)prior distributions $p(\theta)$ to approximate what we know about landslide size distributions so far and irrespective of the data $\mathcal{D}$ studied here.

Our model uses a multi-level setup, in which $i \in \{1, \ldots, n\}$ indexes each landslide observation $x_i$ from a sample of size $n$, and $j \in \{1, \ldots, J\}$ indexes each of $J$ different landslide inventories. The idea of the multi-level model is that the size distribution in each landslide inventory $j$ is characterised by an individual set of GPD parameters $\sigma_j$ and $k_j$. We further assume that the values of each of these inventory-specific parameter pairs are drawn from the same two probability distributions:

$$x_i \sim \mathrm{GPD}(\mu, \sigma_{j[i]}, k_{j[i]}) \tag{3}$$

$$\sigma_j \sim \mathrm{Gamma}(\alpha_\sigma, \beta_\sigma) \tag{4}$$

$$k_j \sim \mathrm{Gamma}(\alpha_k, \beta_k). \tag{5}$$

Here we choose independent Gamma distributions for both $\sigma_j$ and $k_j$ to ensure that the parameters are positive and un-correlated; $\alpha_\sigma$ and $\alpha_k$ are the corresponding shape parameters, and $\beta_\sigma$ and $\beta_k$ are the inverse scale (or rate) parameters. The multi-level model thus learns the parameters for each catalogue informed by both its data, the overarching Gamma distributions, and prior knowledge. While the model allows $\sigma_j$ and $k_j$ to vary between landslide inventories, it also draws on information
from the full data set via this multi-level structure.

Bayesian reasoning requires that we specify our prior knowledge explicitly. We do this by choosing the hyper-parameters of the two Gamma distributions of $\sigma_j$ and $k_j$. These hyper-parameters describe the distribution of landslide scaling parameters across all inventories and offer a global summary from all data. Here, we draw on the growing literature of landslide scaling: recent reviews have summarised that the power-law scaling exponent $\alpha$ for landslide inventories is most often reported in the
150 range of $1 < \alpha < 3$ (Tebbens, 2020). Recalling that the GPD shape parameter $k = 1/\alpha$, we can use this reported range to constrain our (hyper-)prior distributions accordingly. We choose hyper-parameter values such that they contain findings from landslide scaling studies based on data other than the ones used here. The exact shape of these distributions may matter little in the light of the large sample size that informs our likelihood function. We disregard any correlation between the hyper-parameters and simplistically assume independent distributions:

$$\alpha_\sigma \sim \mathcal{N}(1, 0.25) \tag{6}$$

$$\beta_\sigma \sim \mathcal{N}(5, 5) \tag{7}$$

$$\alpha_k \sim \mathcal{N}(6, 1) \tag{8}$$

$$\beta_k \sim \mathcal{N}(9, 1). \tag{9}$$

We assume independent Gaussian distributions for these hyperparameters and choose the prior means and standard devia-
160 tions informed by previous research on landslide scaling properties (Tebbens, 2020).

To avoid having too many inventories with only a handful of large landslides, we consider only those data collections with at least 25 landslides that exceed the threshold size $\mu$. This means that we had to discard many published landslide inventories that only contain smaller slope failures. The data that we need for obtaining the posterior distribution of all GPD parameters consist of the total affected areas by individual landslides and labels of the inventories they belong to. Our data consists of 8627 large
landslides filtered from 37 different inventories (Table 1). Together, these large slope failures affected an area of 6407 km$^2$, or 59% of the total landslide-affected area recorded in these catalogues. The largest landslide is unnamed and extends over 201 km$^2$ in the Caspian Sea basin (Pánek et al., 2016). Our data thus span more than three orders of magnitude in landslide area;

the largest mapped landslide areas per inventory differ by up two orders of magnitude. We note that 19 (or 51%) of our selected landslide inventories were compiled following an earthquake, including five cases of two inventory versions each for the same
event, mapped by different research teams. Only three catalogues (8%) are attributed to a rainfall trigger, while 15 catalogues (41%) are geomorphological inventories that contain information about landslides that accumulated over many years and thus likely reflect various triggers.

The Bayesian implementation of our GPD model requires a numerical approximation of the joint posterior distribution. We use the probabilistic programming language STAN (Carpenter et al., 2017) to code our model and call it via the statistical programming environment **R**. We ran four independent Hamiltonian Monte Carlo chains to explore the model parameter space with the No U-Turn (NUTS) sampler coded in STAN and verified that the numerical solutions converged. Unless stated otherwise, we use medians and 95% highest density intervals (HDIs) to summarise all posterior distributions. A 95% HDI means that there is a 95% probability that a given parameter is in the specified interval (McElreath, 2016).

## 3 Results

### 3.1 Model fits and residuals

We express the size distributions of large landslides in cumulative form using the exceedance probability $p$ for a given landslide area (Figure 2). To measure how well the GPD model fits the data, we compute the residuals in terms of the log-odds ratios between the empirical exceedance probabilities ($p$) and the predicted averages ($\hat{p}$) for each inventory. The log-odds ratio is $\log \frac{\hat{p}(1-p)}{p(1-\hat{p})}$, conditioned on each observed landslide. A positive (negative) log-odds ratio means that the model overestimates (underestimates) the empirical exceedance probability of a given landslide area.

We find that the log-odds ratios reveal most mismatches at either extreme end of the size range, though without any consistency across the inventories (Figure 3). For example, the model underestimates the execeedance probabilities of landslides <0.8 km$^2$ in catalogues M9.1 Tohoku JPN 2 (Tanyaş et al., 2017) and Owyhee USA (Safran et al., 2011), but overestimates the execeedance probabilities of landslides >1.7 km$^2$ in catalogue Dauna Apennines ITA (Ardizzone et al., 2023).

### 3.2 Effects of different landslide inventories

Our model estimates that shape parameters $k_j$ vary across the landslide inventories with posterior medians ranging from $\tilde{k}_j$ = 0.21 in catalogue M8 Haiyuan CHN (Xu et al., 2020) to $\tilde{k}_j$ = 0.92 in catalogue M7.9 Alaska USA 1 (Gorum et al., 2014) (Figure 4). Narrower posterior distributions mean less uncertainty, mainly owing to more large landslides that inform the model in the relevant catalogue. For example, the Campania ITA inventory (Fusco et al., 2023) contains most, i.e. 1854, large landslides, and its 95% HDI is narrowest (0.57 < $k_j$ < 0.72). In contrast, the M6.2 Aisen CHL 1 catalogue (Gorum et al., 2014) has the fewest, i.e. 29, large landslides; its broad posterior distribution is thus informed more by the pooled estimate from all inventories together.

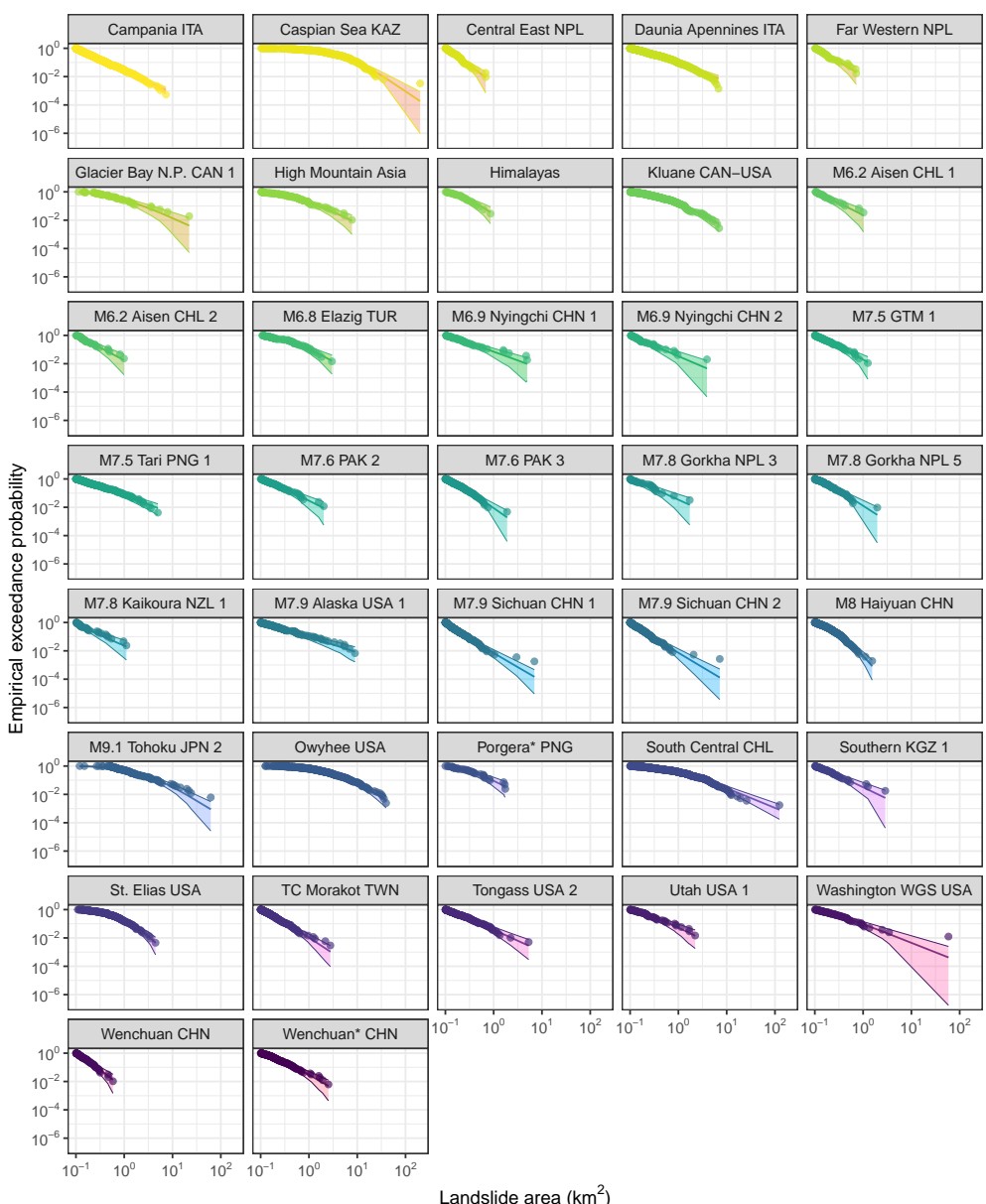

**Figure 2.** Size and frequency of large landslides from 37 inventories that reported at least 25 slope failures affecting $\geq 0.1$ km$^2$ each (Table 1). Circles are observed data ranked by their empirical exceedance probabilities, and lines are posterior medians of a fitted multi-level Generalised Pareto distribution (GPD) with shaded 95% highest density intervals (HDIs). Three-letter codes are ISO country identifiers; numbers refer to different catalogue versions of the same triggering event; catalogues related to earthquakes show estimated magnitude; * denotes inventories derived from deep learning.

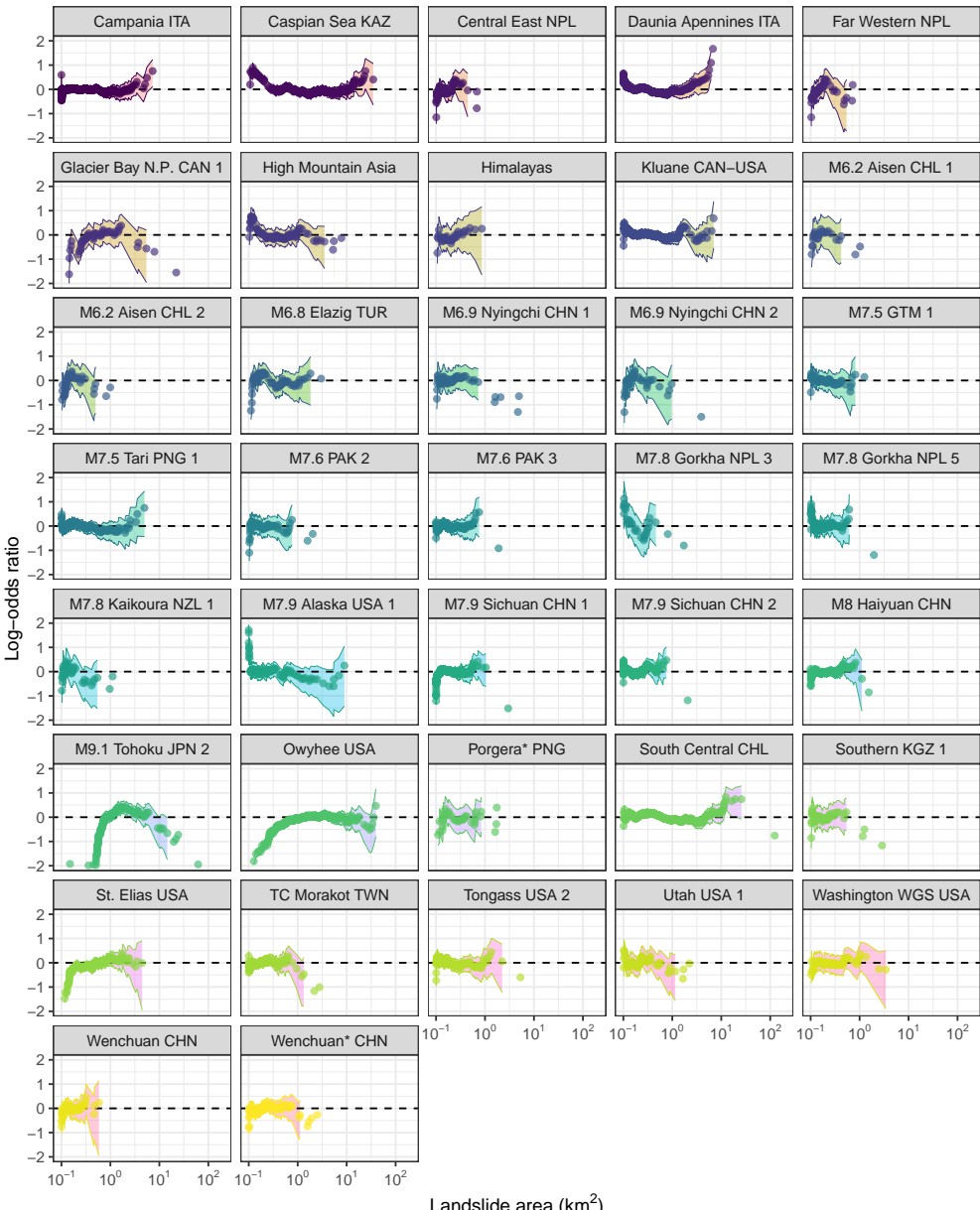

**Figure 3.** Residuals of multi-level GPD fit to large landslide size distributions (Figure 2); residuals are expressed as log-odds ratios of observed versus predicted values. Dashed horizontal lines mark perfect fits; positive (negative) ratios indicate over- (under-) estimated exceedance probabilities. Shaded areas are point-wise 95% HDIs estimated at each landslide observation.

The mean of the Gamma prior distribution is $\bar{k} = \alpha_k/\beta_k$ by definition, and we derive the power-law exponent $\alpha$ from the identity $\alpha = 1/k$. Similarly, we obtain the mean pooled posterior $\bar{\sigma} = \alpha_\sigma/\beta_\sigma$ from the sampled hyperparameters. We find that most of the 95% credible intervals of $k_j$ overlap with that of the mean $\bar{k}$ learned from the pooled model (grey vertical line,

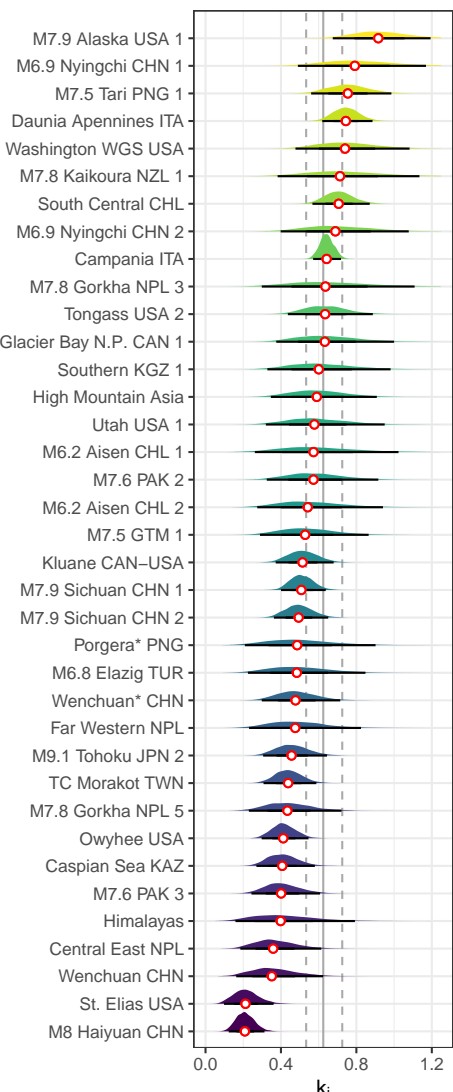

**Figure 4.** Posterior shape parameters $k_j$ of the multi-level GPD model; this parameter is the inverse of the scaling exponent in power-law distributions. White circles are the medians per inventory, and black horizontal lines are 95% HDIs; vertical grey line is the posterior median of the pooled model, and dashed lines delimit its 95% HDI.

flanked by dashed lines marking its 95% HDI). Only two inventories, i.e. one on historic rock avalanches in the St. Elias mountains of Alaska, United States (Bessette-Kirton and Coe, 2020), and one on landslides triggered by the 1920 Haiyuan earthquake, China (Xu et al., 2020), stand out with a $k_j$ that is credibly below that of the population average.

Estimates of $k_j$ differ credibly between inventories in the same geographic region such as western Canada and Alaska, for example when comparing St. Elias USA (Bessette-Kirton and Coe, 2020); Kluane, CAN-USA (W. Smith, pers. comm.); and M7.9 Alaska USA (Gorum et al., 2014). On the contrary, inventories covering very different geographic regions and time

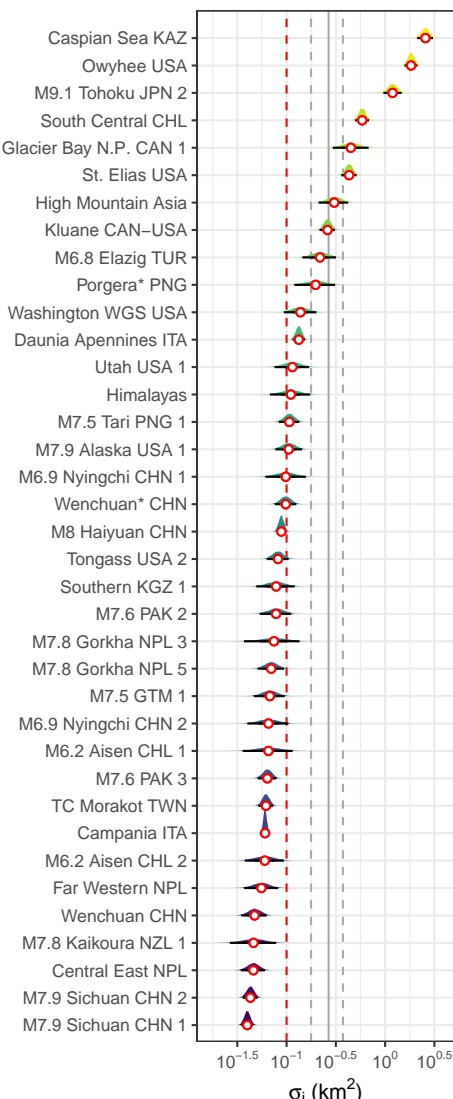

**Figure 5.** Posterior estimates of the GPD scale parameter $\sigma_j$ from the multi-level model. White circles are the medians per inventory, and black horizontal lines are 95% HDIs; vertical grey line is the posterior median of the pooled model, and dashed lines delimit its 95% HDI. Red vertical dashed line is the landslide size threshold $\mu$ fixed at 0.1 km$^2$.

spans have largely overlapping, and thus statistically indifferent, posterior distributions of $k_j$. This is the case, for example, for a catalogue of landslides triggered by the M7.6 Kashmir earthquake in 2005, M7.6 PAK 3 (Basharat et al., 2016), and one on mostly Quaternary landslides in the Caspian Sea basin, Caspian Sea KAZ (Pánek et al., 2016). Similarly, the inventory of
rainfall-triggered landslides in far western Nepal covering 79 time steps between 2002 and 2018, Far Western NPL (Muñoz-Torrero, 2020), and the one on landslides following the 2018 M7.5 Porgera earthquake in Papua New Guinea, a database fully

compiled by a deep learning algorithm, Porgera* PNG (Bhuyan et al., 2023), have indistinguishable posterior distributions of $k_j$. The same goes for inventories mapped by different teams in response to the same earthquake trigger, such as the 2008 Wenchuan earthquake: M7.9 Sichuan CHN 1 (Xu et al., 2014) and M7.9 Sichuan CHN 2 (Li et al., 2014) (Figure 4).

The spread of the posterior scale parameter $\sigma_j$ is more pronounced across the inventories, and the pooled estimate overlaps with those of seven inventories only (Figure 5). Inventory-specific medians range over two orders of magnitude from $\tilde{\sigma}_j = 0.04$ km$^2$ in the M7.9 Sichuan CHN 1 catalogue (Xu et al., 2014) to $\tilde{\sigma}_j = 2.56$ km$^2$ in the Caspian Sea KAZ catalogue (Pánek et al., 2016). Higher values of $\tilde{\sigma}_j$ mark inventories with the more curved fits in Figure 2, especially those to Quaternary landslides such as those of the Caspian Sea, Caspian Sea KAZ (Pánek et al., 2016) or the Columbia River basins, Owyhee USA (Safran

et al., 2011), but also the above-mentioned inventories of rock avalanches that happened in the past few decades (St. Elias USA and Kluane CAN-USA). Again, inventories with different environmental settings and landslide triggers have very similar posterior distributions of $\sigma_j$, such as the one on landslides triggered during Typhoon Morakot, Taiwan, in 2009, TC Morakot TWN (Emberson et al., 2022), and the one on landslides triggered by the M7.6 Kashmir earthquake in 2005, M7.6 PAK 3 (Basharat et al., 2016). Catalogues addressing the same earthquake trigger have largely overlapping posteriors of $\sigma_j$.

## 225 3.3  Pooled estimates of landslide scaling

The pooled estimates in our multi-level model express the variance of the learned parameters across all inventories. The sampled hyperparameters of $k_j$ that describe the shape $\alpha_k$ and rate (or inverse scale) $\beta_k$ of the Gamma-distributed parameter $k_j$ are positively correlated; the same applies for the hyperparameters $\alpha_\sigma$ and $\beta_\sigma$ (Figure 6). From these, we find that the numerical approximation of the joint posterior distribution has a distinct maximum. We obtain a mean power-law exponent of

$1.37 < \bar{\alpha} < 1.85$ across all inventories with 95% probability; the posterior median of $\bar{\alpha}$ is 1.6 (Figure 7). Compared to the prior distribution based on published values of this exponent, our model has gained more certainty from the data considered in this study, yielding a much narrower posterior.

The mean scale parameter is $0.18$ km$^2 < \bar{\sigma} < 0.38$ km$^2$ across all inventories with 95% probability; the posterior median of $\bar{\sigma}$ is $0.27$ km$^2$. This posterior shifted up from the prior distribution that we centered on our arbitrary size threshold for

large landslides. Our model has learned much from the inventory data compared to the priors, especially concerning the high variance of $\sigma_j$ across the individual landslide catalogues.

## 3.4  Comparison with maximum likelihood estimate

To assess how our choice of Bayesian inference aligns with alternative approaches, we compare our results to maximum likelihood estimates (MLE) of the exponent $\alpha_j$ of the inverse power-law distribution, based on the Hill estimator (Clauset

et al., 2009). By definition, the MLE standard error for each inventory decays with the inverse square root of sample size, whereas the Bayesian estimates are informed by all data via the multi-level model structure. Hence we do not expect a 1:1 correspondence from this comparison. Instead, it underlines how variable and uncertain landslide scaling estimates can be for different inventories, regardless of method (Figure 8).

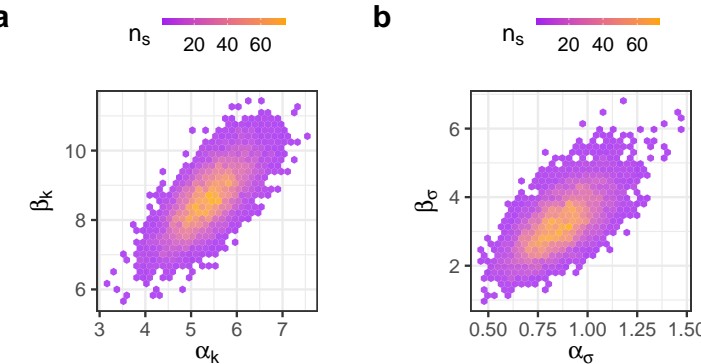

**Figure 6.** Numerical approximations of the hyper-parameter distributions of the GPD multi-level model; sample densities are used to infer probability densities ($n_s$ is sample size). a. Posterior distribution of the hyper-parameters of the Gamma distribution from which the inventory-specific posterior $k_j$ are drawn (Equation 5). b. Posterior distribution of the hyper-parameters of the Gamma distribution from which the inventory-specific posterior of $\sigma_j$ are drawn (Equation 4).

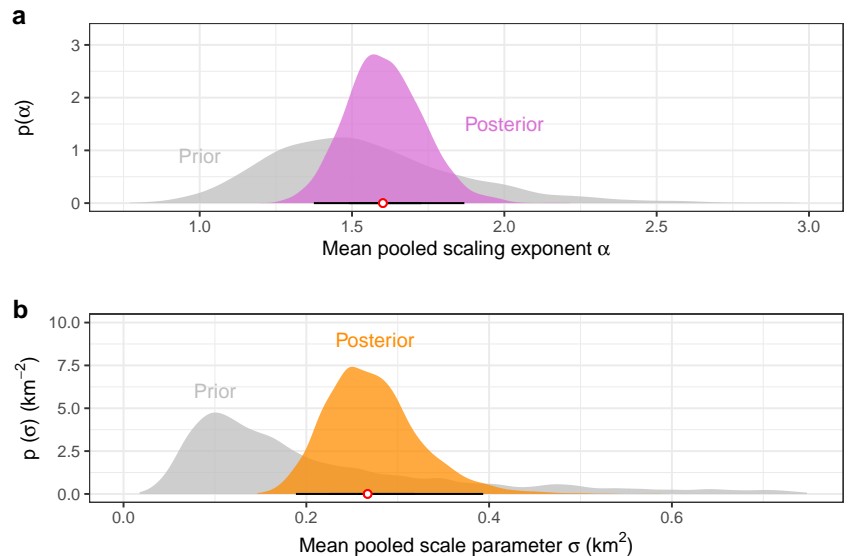

**Figure 7.** a. Prior and posterior distributions of inverse power-law scaling exponent $\alpha$ for the pooled model across all landslide inventories. White circle is pooled posterior median and black horizontal line is 95% HDI. b. Prior and posterior distributions of GPD scale parameter $\sigma$ for the pooled model. White circle is pooled posterior median and black horizontal line is 95% HDI.

We obtain inventory-specific MLEs of $0.33 < \hat{\alpha}_j < 2.39$. This spread encompasses most reported values in the literature (Tebbens, 2020). In contrast, the posterior medians of $\alpha_j = 1/k_j$ occupy a seemingly broader range ($1.09 < \tilde{\alpha} < 4.81$), though nominally similar to that of the MLE method for most inventories within the respective errors. However, the coefficient of variation is narrower for the Bayesian median estimates. Two inventories stand out with very high scaling exponents, i.e. M8

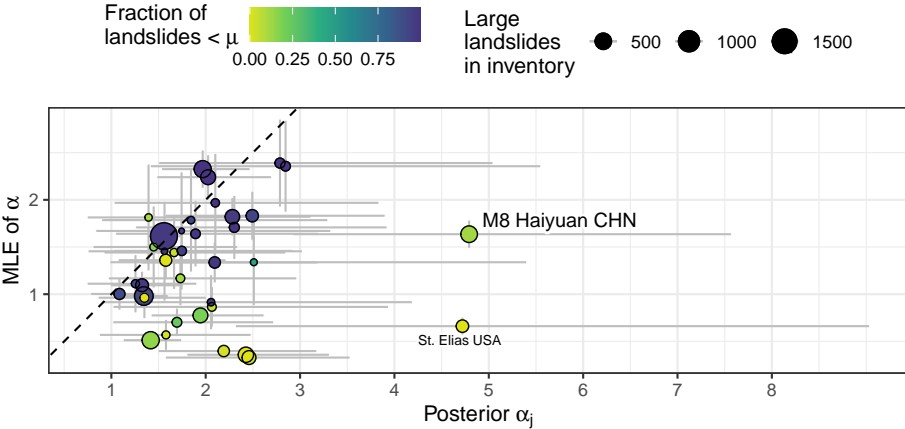

**Figure 8.** Comparison between posterior GPD estimates and the maximum likelihood estimator (MLE) of scaling exponents $\alpha$ for each inventory. Dashed 1:1 line is for visual comparison only. Colour scale shows the fraction of (discarded) landslides below the size threshold $\mu$ in each inventory; bubbles are scaled to sample size of large landslides used for parameter estimation. Grey vertical bars span two standard errors about the mean; grey horizontal bars are 95% HDIs. Axes are scaled equally.

Haiyuan CHN and St. Elias USA. Both inventories have only few landslides that we censored because they were below the size threshold $\mu = 0.1$ km$^2$; in other words, these catalogues contained mostly large landslides originally.

## 4    Discussion

### 4.1    Implications

We offer estimates of scaling statistics that characterise the size distribution of rare, large landslides ($\geq 0.1$ km$^2$), informed by thousands of data points from dozens of inventories, and findings from previous research. Instead of extrapolating models fit to mainly smaller landslides, we use a dedicated peak-over-threshold approach that uses observations on large landslides exclusively. The narrowest 95% HDI of $\alpha_j$, and thus the best we can constrain this parameter, is that of the Campania ITA catalogue with $1.39 < \alpha_j < 1.74$. This numerical range has much overlap with that of previously reported scaling exponents that were obtained for mostly smaller landslides, though (Tebbens, 2020). Still, the nearly triangular posterior distribution has much of its probability mass near its peak (Figure 4), and the same goes for the pooled estimate (Figure 7a). Except for two cases, the posterior $\alpha_j$ for landslide inventories remain indistinguishable from the pooled estimate. This low variance of $\alpha_j$ across inventories is striking if we consider the diverse mapping techniques, levels of data quality, coverage, environmental setting, and landslide triggers. The inventories we selected cover several climatic zones with different vegetation and land cover characteristics that likely affect the preservation, detection, and mapping of evidence of large landslides. Moreover, some inventories were generated from deep learning algorithms (Bhuyan et al., 2023), whereas most others were mapped manually. While many posterior estimates of $\sigma_j$ are close to our arbitrary size threshold for large landslides of 0.1 km$^2$, the variance in

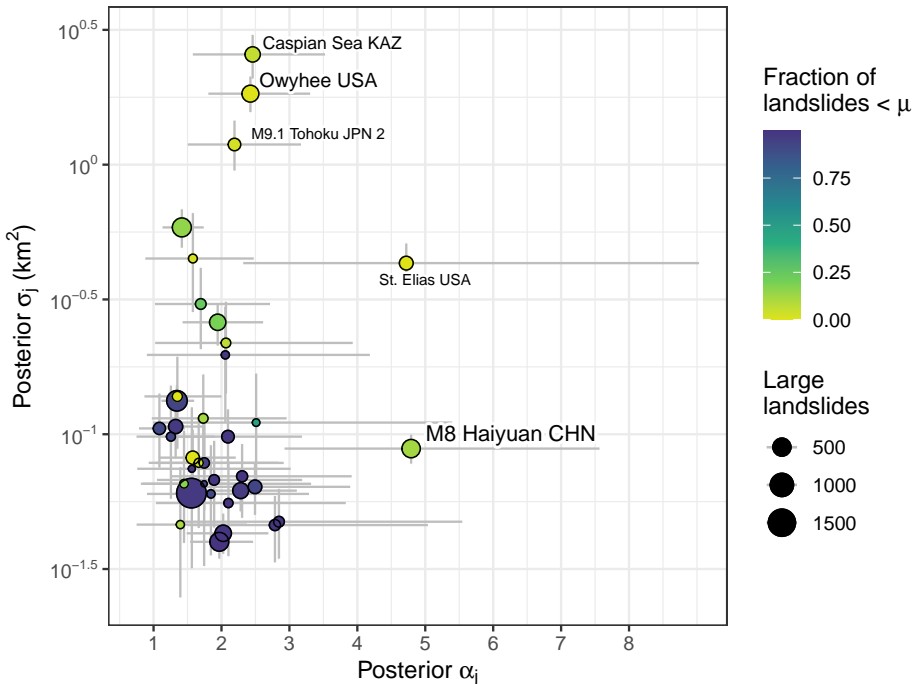

**Figure 9.** Median posterior parameter estimates as a function of the number of large landslides per inventory (bubble size) and the fraction of discarded landslides below the size threshold (colour scale). Grey bars span the 95% HDIs.

this parameter is high compared to the pooled estimate. Overall, the median estimates of both $\alpha_j$ and $\sigma_j$ are unaffected by the number of large landslides reported in a given inventory (Figure 9).

Our model highlights several inventories with scaling statistics that stand out. The M8 Haiyuan CHN and St. Elias USA catalogues have high estimates of $\alpha_j$, while Caspian Sea KAZ, M9.1 Tohoku JPN 2, and Owyhee USA have high estimates of $\sigma_j$ well beyond the central tendency of most other catalogues (Figure 9). These inventories consist almost exclusively of large landslides and have much fewer landslides below the size threshold $\mu$. In contrast, we observe that inventories with most landslides falling below $\mu$ have low values of $\sigma_j$ consistently. We infer that landslide catalogues that were focused on compiling information about larger landslides tend to have elevated values of $\sigma_j$. In this context, our selection of inventories is nearly balanced: twenty of them have less than 10% large landslides, whereas seventeen of them have more than 50%. One possible explanation for these outlying landslide-size statistics is that the GPD is a poor fit to inventories that mainly feature large landslides, at least for the chosen threshold $\mu$. The residuals for most of these inventories show pronounced underestimates for the smallest landslide range, except for the South Central CHL and Caspian Sea KAZ catalogues (Figure 3). Yet, other inventories with similar residuals (e.g. Glacier Bay N.P. CAN) hardly stand out compared to the pooled estimates. Clearly, extrapolating the model across the full size range, for example to infer the number of seemingly missing, underreported, or overlooked landslides (Tanyaş et al., 2019) can be misleading in these cases. Another explanation for the high estimates of $k_j$ and $\sigma_j$ is that the original mapping was focused on landslides close to, or well above, our choice of $\mu = 0.1$ km$^2$, such that

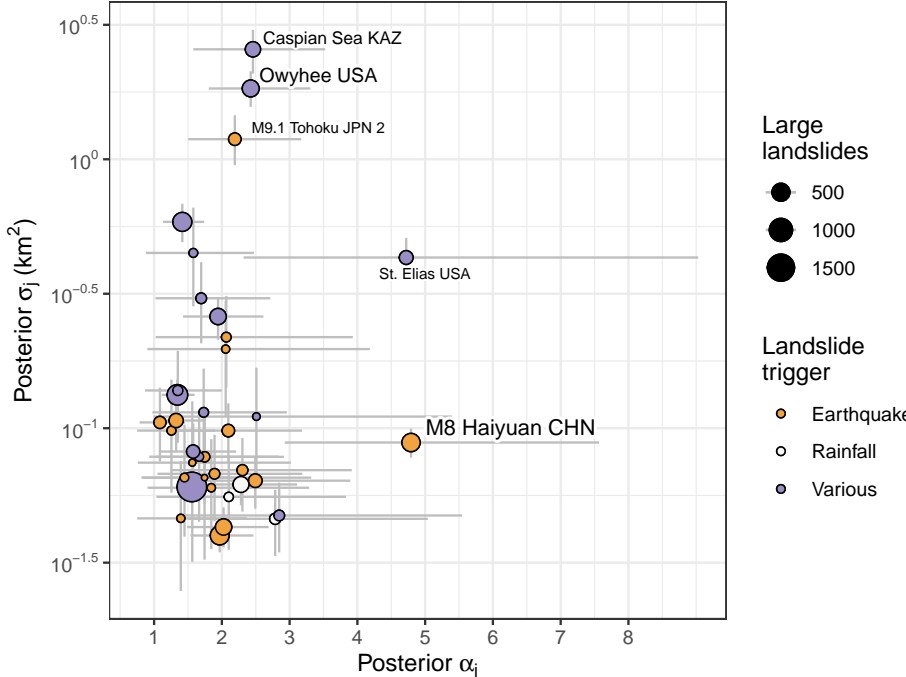

**Figure 10.** Median posterior parameter estimates as a function of sample size (bubble size) and type of trigger (colour scale).

undersampling of landslides near this size threshold may explain some of the variance of scaling estimates. Reconstructing historic landslide episodes from old air photos and preserved geomorphic evidence (e.g. M8 Haiyuan CHN) may also add variance. Either way, the strategy for keeping such mapping practical is to use a size cutoff. Hence, although smaller landslides may be recognised, they are excluded and thus censored in these inventories. Some of these inventories also contain partly

overlapping slope-failure deposits of multiple ages, marking several phases of reactivation. Such overlaps may cause more landslides to surpass the size threshold. Hence, the mapping objective would partly bias estimates of $\sigma_j$ for a size threshold that is too low.

We also find that the 95% credible intervals of both GPD parameters overlap for landslide inventories regardless of whether they were attributed to recent earthquakes and rainstorms, or whether they integrate landslide observations, and thus likely

various triggers, over many years (Figure 10). We infer that the scaling statistics disclose very little about the type of landslide trigger. In this context, our findings caution against a mechanistic interpretation of scaling parameters, at least for large slope failures. These can have longer and more complex histories of precursory slope deformation and failure than smaller landslides (Korup et al., 2007), and likely respond to stresses that accumulate over repeated episodes of earthquake shaking or rainstorms.

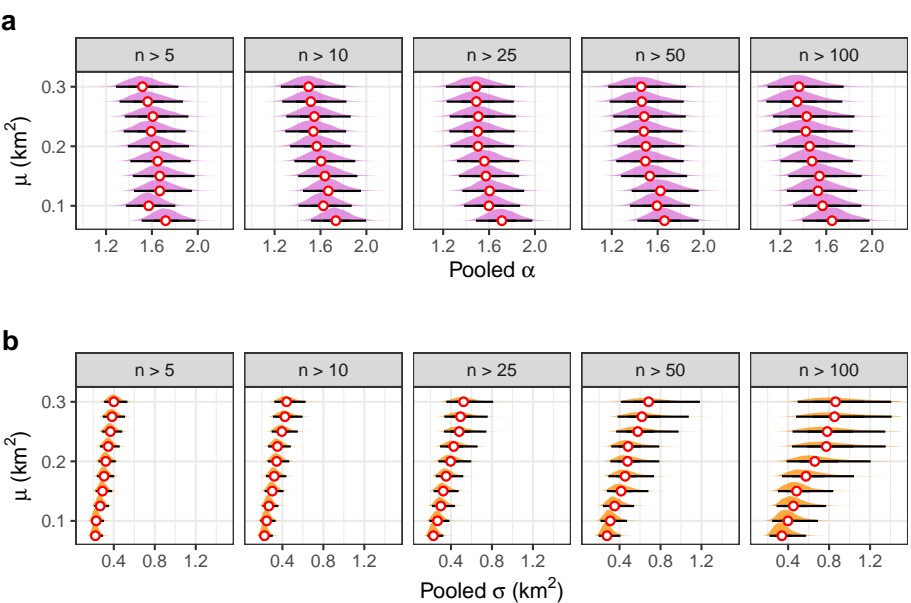

**Figure 11.** Effect of varying size threshold $\mu$ and minimum sample size $n$ in each inventory on posterior estimates of the pooled scaling exponent $\alpha$ and the pooled parameter $\sigma$. White bubbles are posterior medians, and horizontal black bars delimit the 95% HDIs.

## 4.2 Role of size threshold and sample size

The heavy-tailed distribution of landslides means that we discarded many samples for small increases in $\mu$. To test the sensitivity of our results to the choice of size threshold $\mu$, we replicated our analyses, and recorded the variation of the pooled estimates $\bar{\alpha}$ and $\bar{\sigma}$ as a function of both $\mu$ and the minimum number of large landslides that an inventory needs to have to be included in our model. We find that varying the size threshold such that $0.075\ \text{km}^2 < \mu < 0.3\ \text{km}^2$ returns posterior pooled values of $\alpha$ that decrease slightly with increasing $\mu$ and the minimum number of samples per inventory, though with much overlap (Figure 11).

Overall, $1.23 < \bar{\alpha} < 1.9$ with 95% probability regardless of the threshold or sample size that we pick. Estimates of $\bar{\sigma}$ increase sightly with $\mu$, but also with some overlap. Preferring larger inventories for a given size threshold reduces the total sample size such that the pooled posterior distributions of $\sigma$ get broader.

   We infer that the choice of $\mu$, and hence the definition of "large" landslides (McColl and Cook, 2024), has limited influence on the scaling statistics, and especially $\bar{\alpha}$. Values of $\mu$ below the range that we tested violate the assumption of a high threshold

such that a GPD would be inappropriate, whereas values above this range suffer from too small sample sizes. The pooled scale parameter $\bar{\sigma}$ shows less variation with $\mu$ and has largely overlapping posteriors. Hence, our choice of the minimum number of large landslides per inventory ($n = 25$) limits both the number of groups and the overall sample size in our model. Admitting more inventories that contain fewer large landslides changes the posterior $\bar{k}$ and $\bar{\sigma}$ only slightly, but does narrow the uncertainties, especially for higher thresholds $\mu$.

## 4.3 Benefits

Our Bayesian multi-level approach expands on previous, though largely separate, efforts of comparing landslide size statistics across different inventories (Tebbens, 2020). We offer here a single, consistent model that has several benefits:

First, the Bayesian setup can handle the small sample problem of large landslides. Scaling parameters for large landslides from a single landslide inventory are commonly estimated from extrapolating model fits that largely draw on more numerous, smaller landslides. Yet, even simulated power-law distributed data have natural scatter in the largest of observations (Clauset et al., 2009), making it difficult to validate extrapolations and leading to over-confident parameter estimates, especially when ignoring the attached errors (Figure 8). Using data from other landslide inventories to validate these estimates tacitly assumes that the scaling parameters have similar errors, but offers no way of determining of whether this assumption is at all valid. The multi-level model instead draws on the larger sample size from all inventories, and explicitly refines this shared knowledge in dedicated posterior distributions for each catalogue. These group-level parameter estimates tend to be closer to the pooled mean than those derived for separate models using fewer data from each group alone. This effect is known as parameter shrinkage (McElreath, 2016) and guards against overfitting, especially for inventories with few data.

Second, the Bayesian treatment quantifies all parameter uncertainties explicitly, and especially those that capture previous knowledge about landslide-size distributions. We can thus quantify how much we have learned from the data by comparing the posterior and prior distributions (Figure 7). By design, a Bayesian model seeks a compromise between these previous findings and the data considered here in a probabilistically consistent way. To this end, we made sure to include mostly recently published landslide inventories or those that had not been considered in scaling studies before.

Third, the multi-level model structure enables direct comparison of parameter estimates across and between landslide inventories (Figure 10). Any differences in the underlying workflows of detecting and mapping landslides and the commensurate sample sizes are being accounted for by separate posterior distributions and their deviation from the pooled estimates (Figures 4, 5). Our model measures objectively how similar landslide inventories are in terms of the scaling parameters that jointly, instead of separately, define the size distributions of large landslides.

Fourth, the peak-over-threshold approach that defines the GPD is rooted in extreme-value theory and thus expresses what we can expect statistically from the size distribution of large landslides. The parameters of the GPD contain information about the power-law scaling, and translate readily into parameters of other distributions used to characterise the size scaling of landslides. Given that most inventories focused on large landslides have to operate on a lower size threshold, we argue that the GPD is a natural choice for characterising the size distributions of more extreme slope failures.

Finally, we can flexibly modify our multi-level model in several ways. One option is to group the data in other ways than by inventories. For example, we can specify the group levels such that they represent dominant landslide, soil, or rock type; or any other characteristic that may have been collected during the process of compiling the landslide inventory. We discarded the option of using the type of landslide trigger as a group level because we only have three inventories of rainfall-triggered landslides, so that posterior estimates of scaling parameters might rely too much on the more numerous data in earthquake-triggered and multi-event catalogues. Adding inventory type as yet another group level would expand the parameter space

and unnecessarily add bias for multi-event catalogues that are likely dominated by an unknown fraction of either rainfall- or earthquake-triggered landslides. Instead, our choice of priors remains impartial to inventory type. We recall that the GPD is by definition "blind" to data below the threshold $\mu$ in that it truncates all observations below the threshold. One alternative is to also directly learn $\mu$ from the data, either globally or per inventory, and add further covariates that may control the form of the GPD.

## 5  Conclusions

We propose a multi-level model as common ground for consistently estimating and comparing size distributions of large and rarely observed slope failures across different inventories. In choosing a peak-over-threshold approach, the Generalised Pareto distribution (GPD) reflects what we would expect statistically from a given landslide size distribution. The multi-level setup remediates the problem of low sample size by making use of all available data for estimating scaling parameters while acknowledging inherent differences across inventories. Our model results based on 37 inventories with 8627 large landslides ($\geq 0.1$ km$^2$) show that the power-law exponent for each inventory $\alpha_j = 1/k_j$ discloses little about the different underlying landslide trigger(s), geographic region, or time span concerning a given inventory, i.e. whether it is event-based or compiles landslides of many different ages. Inventories of mostly undated, prehistoric landslides have scaling exponents $\alpha_j$ that hardly differ from those of historic, event-based catalogues. While several studies have attributed a physical meaning to scaling statistics of landslide size, we argue that some of this meaning might get diluted or even lost in empirical data that may combine confounding controls. We surmise that landslide inventories record these physical processes, though in a mixed way that could admit, for example, different failure types, rock and soil types, and groundwater conditions. We suspect that most landslide inventories have mixed size distributions. For example, mixing data from inventories with differing size thresholds could add variance to $\sigma_j$. At least for the large landslides studied here, the scaling statistics likely reflect bulk physical characteristics instead of variables of a single deterministic model of slope stability. Despite thousands of large landslides to learn from, the uncertainty about $\alpha_j$ spans several decimal points. If taking all inventories together, the pooled $\alpha$ captures most of this variance. The GPD scale parameter $\sigma_j$ has more spread across inventories and is affected especially by those with few or no landslides below our size threshold. We infer that $\sigma_j$ is sensitive to the desired landslide size range and likely reflects the influence of mapping choices, and specifically the compromise of finding a suitable size threshold. Regardless, the choice of probability distribution used to model landslide areas is arbitrary, and parameter estimates disclose nothing about sample size or completeness. We advise against inferring any completeness from the GPD or any other distribution because probability densities are conditional on a model, and models should be fitted to data and not vice versa. Finally, our model measures objectively by how much scaling statistics differ across inventories within estimation error. Such differences can be vital if using scaling statistics for predicting future landslide hazard in terms of size and frequency (Hergarten, 2023).

*Code and data availability.* We used the statistical programming environment **R** (https://cran.r-project.org) with RStudio (https://posit.co) as a frontend for all data analysis, and coded the Bayesian GPD in the probabilistic language STAN (https://mc-stan.org); all this software is freely available. The landslide inventories studied here have either been published as indicated. All code needed to reproduce this analysis is available in **R** notebook format upon request.

*Author contributions.* O.K. designed the study, carried out the data pre-processing and analyses, and wrote the manuscript. L.V.L. and J.V.F. helped with the collection and compilation of landslide inventories. All authors contributed equally to editing the final version of the manuscript.

*Competing interests.* The authors declare that they have no competing interests.

*Acknowledgements.* Will Smith and Stuart Dunning kindly shared data on landslides in Kluane National Park. Part of this research was funded by the German Research Foundation via the Research Training Group NatRiskChange (DFG GRK 2043).

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

**Table 1.** List of 37 landslide inventories with sample size of landslides $\geq 0.1$ km$^2$; largest and smallest landslide reported; count of discarded landslides below $\mu$; and trigger. Three-letter codes are ISO country identifiers; numbers refer to different catalogue versions of the same triggering event; catalogues related to earthquakes show estimated magnitude; * denotes inventories derived from deep learning.

| Inventory | Samples | Max. area (km$^2$) | Min. area (km$^2$) | $n$ below $\mu$ | Trigger | Reference |
|---|---|---|---|---|---|---|
| Campania ITA | 1854 | 7.3 | 0.0000 | 79362 | Various | Fusco et al. (2023) |
| Caspian Sea KAZ | 303 | 201.1 | 0.0170 | 18 | Various | Pánek et al. (2016) |
| Central East NPL | 108 | 0.7 | 0.0003 | 12730 | Rainfall | Jones et al. (2021) |
| Daunia Apennines ITA | 712 | 6.8 | 0.0001 | 10091 | Various | Ardizzone et al. (2023) |
| Far Western NPL | 58 | 0.7 | 0.0000 | 26292 | Rainfall | Muñoz-Torrero (2020) |
| Glacier Bay N.P. CAN 1 | 52 | 22.2 | 0.0972 | 1 | Various | Kim et al. (2022) |
| High Mountain Asia | 96 | 7.8 | 0.0155 | 31 | Various | Liu et al. (2021) |
| Himalayas | 35 | 0.9 | 0.0219 | 32 | Various | Marc et al. (2019) |
| Kluane CAN-USA | 369 | 7.0 | 0.0190 | 89 | Various | Smith et al. (2023) |
| M6.2 Aisen CHL 1 | 29 | 1.0 | 0.0004 | 488 | Earthquake | Gorum et al. (2014) |
| M6.2 Aisen CHL 2 | 43 | 1.0 | 0.0002 | 495 | Earthquake | Sepúlveda et al. (2010) |
| M6.8 Elazig TUR | 68 | 3.0 | 0.0905 | 5 | Earthquake | Karakas et al. (2021) |
| M6.9 Nyingchi CHN 1 | 54 | 4.9 | 0.0001 | 712 | Earthquake | Hu et al. (2019) |
| M6.9 Nyingchi CHN 2 | 49 | 3.9 | 0.0914 | 10 | Earthquake | Zhao et al. (2019) |
| M7.5 GTM 1 | 90 | 1.3 | 0.0000 | 6134 | Earthquake | Harp et al. (1981) |
| M7.5 Tari PNG 1 | 239 | 5.0 | 0.0001 | 11369 | Earthquake | Tanyaş et al. (2022b) |
| M7.6 PAK 2 | 84 | 2.0 | 0.0000 | 1369 | Earthquake | Basharat et al. (2014) |
| M7.6 PAK 3 | 209 | 1.9 | 0.0000 | 2721 | Earthquake | Basharat et al. (2016) |
| M7.8 Gorkha NPL 3 | 31 | 1.7 | 0.0000 | 24884 | Earthquake | Roback et al. (2018) |
| M7.8 Gorkha NPL 5 | 105 | 2.0 | 0.0000 | 21046 | Earthquake | Valagussa et al. (2021) |
| M7.8 Kaikoura NZL 1 | 42 | 1.1 | 0.0920 | 5 | Earthquake | Tanyaş et al. (2022a) |
| M7.9 Alaska USA 1 | 147 | 9.0 | 0.0009 | 1432 | Earthquake | Gorum et al. (2014) |
| M7.9 Sichuan CHN 1 | 556 | 7.0 | 0.0000 | 196925 | Earthquake | Xu et al. (2014) |
| M7.9 Sichuan CHN 2 | 373 | 7.2 | 0.0000 | 69233 | Earthquake | Li et al. (2014) |
| M8 Haiyuan CHN | 519 | 1.5 | 0.0902 | 70 | Earthquake | Xu et al. (2020) |
| M9.1 Tohoku JPN 2 | 162 | 62.0 | 0.0600 | 4 | Earthquake | Tanyaş et al. (2017) |
| Owyhee USA | 412 | 39.6 | 0.1253 | 0 | Various | Safran et al. (2011) |
| Porgera* PNG | 41 | 1.7 | 0.0002 | 1516 | Earthquake | Bhuyan et al. (2023) |
| South Central CHL | 571 | 123.9 | 0.0116 | 101 | Various | Antinao and Gosse (2009) |
| Southern KGZ 1 | 56 | 2.9 | 0.0905 | 3 | Various | Behling et al. (2016) |
| St. Elias USA | 220 | 4.5 | 0.1096 | 0 | Various | Belair et al. (2022) |
| TC Morakot TWN | 340 | 2.8 | 0.0000 | 9896 | Rainfall | Emberson et al. (2022) |
| Tongass USA 2 | 195 | 5.3 | 0.1002 | 0 | Various | Belair et al. (2022) |
| Utah USA 1 | 68 | 2.2 | 0.0926 | 9 | Various | Belair et al. (2022) |
| Washington WGS USA | 80 | 58.6 | 0.1012 | 0 | Various | Belair et al. (2022) |
| Wenchuan CHN | 96 | 0.6 | 0.0000 | 9933 | Various | Domènech et al. (2018) |