# Peer review of "Size scaling of large landslides from incomplete inventories"

_Natural Hazards and Earth System Sciences, 2024_

## Referee Comment (RC1)

[referee-annotated manuscript omitted]

---

## Author Comment (AC2)

Discussion of Preprint NHESS-2024-55 (https://doi.org/10.5194/nhess-2024-55)
**Size scaling of large landslides from incomplete inventories**
Oliver Korup, Lisa Luna, and Joaquin Ferrer

AUTHORS' REPLY TO REVIEWERS

Dear Reviewers, dear Editors,

We thank both anonymous reviewers for their constructive comments and positive appraisals of our study. Below we offer a point-by-point response (marked as ***paragraphs in bold italics, beginning with "A:"***) to these comments, and how we plan to address these suggestions consistent with the recommended minor revisions. All line, figure, and table numbers refer to the original manuscript.

Best wishes,

Oliver Korup
On behalf of all co-authors

RC1: 'Comment on nhess-2024-55', Anonymous Referee #1, 24 May 2024
Overall, the paper presents a new approach to model the landslide size distribution of the so defined "large landslides", setting a threshold at $10^5$ m$^2$. The described approach is rooted in the Bayesian framework and builds the estimation of the posterior probability distributions on previous knowledge (literature) and on several inventories that were publicly available. The approach presented builds the prior distributions using all the landslides available in the inventories, without distinguishing, by choice, among different inventories (event, multi-temporal, geomorphological), triggers, type of failures, experience of the mappers, method of mapping. The authors claim that this method proved that the power-law scaling of large landslides is not purely mechanistic, as results show no difference of statistics among a quite wide set of landslides and inventories.
The paper is written in very good English, concepts and ideas are clearly expressed, with a sound logical consequentiality. The effect is that it is easy to read and understand. Figures are good looking, self-explanatory and consistent, well captioned and properly cited in the text. There are no missing nor unnecessary figures.
The Introduction is well-framed, and the research niche is clearly defined. Methods are accurately and clearly described, both the theoretical framework and the practical aspects, as well as the working assumptions and their legacy. Results are clearly presented and illustrated and, in general, the discussion highlights the main aspects related to the research questions posed.

***A: We thank the reviewer for these kind words and the positive assessment of our work.***

I have a few questions/concerns that need to ask the authors. The rest of my few comments can be found in the annotated pdf.

The first question is about the incompleteness of large landslides in inventories. In the Introduction, it is stated that (lines 42-47) "*Still, most uncertainty remains about the large landslides that are rarely sampled. … One reason for this knowledge gap is that large landslides are often elusive in catalogues compiled shortly after a landslide-triggering earthquake or rainstorm (Hao et al., 2020; Abancó et al., 2021; Santangelo et al., 2023). Sample sizes often involve only a handful to several dozen large landslides, and thus often remain too small for robust statistics in a given study area. Hence, inference is mostly based on the simple extrapolation of model fits beyond the observed size range.*" In my experience as geomorphologist involved in several landslide mapping activities, when preparing landslide inventory maps, the source of incompleteness of inventory maps is predominantly due to missing small landslides, which are the most elusive. If an event occurs and landslides are smaller than 0.1 km$^2$, that does not necessarily mean that large landslides were under sampled, but it cannot be excluded to be a peculiar feature of that specific event, which was the case of the event in the Marche region, cited in this paper (Santangelo et al., 2023). I think in the end the problem this paper is facing does not really change for this, as the sample of large landslides is often limited in the inventories because their number is, as a matter of fact, far smaller than smaller

landslides. What I have seen often, is that many geomorphological historical inventories (sensu Malamud et al., 2004; Guzzetti et al., 2012) lack very large and very dismantled landslides, where the evidence needed to recognise and map them is more complex, and often requires higher geological skills, long and wide experience and, above all, time and dedication. Also, I do not understand the reference in the title to "incomplete inventories". I do not think I get this. I would just refer to inventories in general.

*A: We agree that any potential undersampling of large landslides is independent of the number of landslides < 0.1 km² or any other size threshold; this is why we checked whether our scaling estimates are sensitive to varying thresholds (Figure 11). We also concur that large landslides can elude inventories, especially if these are compiled for rapid response or a focus on fresh evidence (lines 69-70). This possible lack of large landslides motivates our use of "incomplete inventories" instead of simply "inventories". We will add to the introductory text that "Landslide inventories can be incomplete in that they miss out on those large landslides that have indistinct or less obvious geomorphic evidence, and thus require experience, skill, and time for accurate detection and mapping". We believe that keeping "incomplete" reflects how, in some inventories, we might miss out on evidence of larger slope failures.*

The second question is about the effect of building the priors by putting all landslides together. I am wondering what the effect would be of estimating the posteriors building on priors only based on specific types of landslide inventories. So, treating separately different types of inventories, event-based (and multi-event), geomorphological, and multi-temporal. Would the estimates be similar or different, and what would be the interpretation of that result? As the authors correctly checked the choice of the size threshold, in my opinion this one also needs to be checked and commented.

*A: We agree that it would be interesting to specify different priors for each type of landslide inventory, thus distinguishing between rainfall- vs. earthquake-triggered or multi-event inventories ("trigger type" in Figure 10). Yet, some of this variation is already captured in the hyperparameters of our model, which describe the variance of scaling parameters across inventories (Figures 6, 7). Technically, however, we can only specify priors for model parameters and levels. We could admit the type of inventory as a group level to our hierarchical model, but discard this option for two reasons: (1) we only have three inventories of rainfall-triggered landslides (line 168), so that posterior estimates of scaling parameters might rely too much on the more numerous data in earthquake-triggered and multi-event inventories; (2) adding inventory type as yet another group level would expand the parameter space and unnecessarily add bias for multi-event catalogues that are likely dominated by an unknown fraction of either rainfall- or earthquake-triggered landslides. Instead, our choice of priors remains impartial to inventory type.*

The third question is about using this kind of landslide size statistics as a tool for inferring the degree of completeness of inventories before looking at the data. This could be added as a topic in the discussion section.

*A: We will add the following statement to the discussion: "Regardless, the choice of probability distributions used to model landslide areas is arbitrary, and parameter estimates disclose nothing about sample size or completeness. We advise against inferring any completeness from these or any other distributions because probability densities describe relative frequencies, and as models should be fitted to data and not vice versa." A landslide inventory is "substantially complete" (Malamud et al., 2004; line 459) if nearly all landslides that had happened were detected and mapped eventually. Completeness thus qualifies the observation and recording process of the data. Incomplete inventories might be regarded as samples, if assuming that all landslides that were triggered by an earthquake or rainstorm, for example, form a statistical population. This assumption may need to be relaxed for multi-event inventories, however, as defining their "completeness" requires a specific area and time interval. In our case, we find it prudent to assume incomplete inventories, given the rarity of large landslides above an arbitrary size threshold.*

Minor comments and typos can be found in the annotated pdf.
In conclusion, I think the paper is suitable for publication in NHESS pending minor revisions.
Best regards.

*Annotated PDF (keyed to line numbers):*

Figure 1: add geomorphological historical inventories

*A: Figure 1 is concerned with data and models in general, so "geomorphological historical inventories" would be part of the "data" here, as would be any other type of catalogue. We will underline in the figure caption.*

59: the methods to detect and map landslides and to compile landslide inventories…

*A: Yes, we will add "inventories" here.*

70: "or simply those that are easiest to recognise" - not sure this is a correct consequence of experience. That would mostly highlight limited experience.

*A: True, although even experienced researchers may map large (and older) landslides differently (van Den Eeckhaut et al., 2005; line 67 in our original manuscript). We will add "in case of limited experience" here.*

169: "accumulated over many years and thus likely reflect various triggers." - i.e. geomorphological historical inventories?

*A: We will insert "geomorphological inventories" here, but avoid using "historical", as these catalogues rarely have time-stamped or otherwise dated landslides.*

254: "however" - you meant: though?

*A: Yes, we will change this.*

324: "4, 4" - Perhaps it is figures 4 and 5 here

*A: Yes, this should read "Figures 4, 5".*

RC2: 'Comment on nhess-2024-55', Anonymous Referee #2, 28 Aug 2024

Dear authors,

I provide the referee comments to the manuscript entitled ''Size scaling of large landslides from incomplete inventories''.

General comments:

The manuscript is clearly presented and well-structured, making it easy to read and follow. The addressed topic is relevant to the field of landslide research, particularly of landslide inventory mapping and statistical modelling of landslide inventories. The methods are clearly outlined and described with enough detail. The English is high level, contributing to the clarity of the work.

Finally, the manuscript is prepared according to the NHESS journal's standards and can be published after minor revision.

*A: We thank the reviewer for this positive appraisal of our work.*

Below, I provide a few specific comments.

Comment 1:

I suggest improving the presentation of the landslide inventories used in the study. I recommend adding this relevant information about the landslide inventories to Table 1: type of the inventory; method for its preparation; expert team, if possible; and area covered. I believe that including this information would enhance the understanding of the reliability of the input data. Moreover, since the concept of incomplete inventories plays an important role and is included in the title, it would be useful to briefly clarify the main reasons for their incompleteness.

*A: Please note that inventory type is already featured as "Trigger" (Table 1). While the method of compilation is of interest, we surmise that even the same method (e.g. mapping from air photos) can yield different results for different study areas (given varying image qualities, cloud cover, shadow, etc.). Hence, we used the landslide inventory as the most immediate level of grouping the landslide data. We did, however, outline catalogues that were derived from deep learning with an asterisk (\*). The composition of the expert team is difficult to reproduce and assess, and we are unsure how disclosing this would "enhance the understanding of the reliability of the input data". The same goes for the area covered, which is rarely, let alone consistently, reported. Hence, adding estimates of areas covered for each inventory would elevate, instead of reduce, uncertainties. We recall that the reliability of the input data concerns, solely, the mapped landslide areas; the size of study area does not enter the model.*

*We had outlined the main reasons for incompleteness of landslide inventories in lines 59-73. Limited or more difficult recognition of large landslides, especially if enlarged, eroded, or partly buried, appears to be one of the major reasons. Reviewer 1 agrees by stating that "What I have seen often, is that many geomorphological historical inventories (sensu Malamud et al., 2004; Guzzetti et al., 2012) lack very large and very dismantled landslides, where the evidence needed to recognise and map them is more complex, and often requires higher geological skills, extensive experience and, above all, time and dedication." We will add this accordingly (please see our reply to reviewer 1 above).*

Comment 2:
A significant number of references on the landslide inventories, cited in the Table 1, are not included in the References list. Please, thoroughly review the citations and add any missing references to the list.

*A: We thank the reviewer for picking this up. A number of references in the original LaTeX source file failed to compile and appear in the final PDF. This has now been fixed.*

Comment 3:
The statement in the Abstract (line 13) "Our model identifies several inventories with outlier scaling statistics that reflect intentional censoring during mapping" raises some important concerns. While this is a significant assertion, it would be more compelling if it were supported by concrete evidence or detailed analysis in the Results or Discussion section. Without sufficient evidence, the claim appears speculative. I recommend providing more justification or proof that clearly demonstrates the intentional censoring you refer to, ensuring that the statement is well-founded and substantiated.

*R: We will add the following explanation to the discussion: "Figures 9 and 10 highlight (and label) several inventories with posterior scaling parameters $k_j$ and $\sigma_j$ well beyond the central tendency of most other catalogues. We note that these outlier inventories have much fewer landslides below the size threshold (yellow-green bubbles, Figure 9), and surmise that their main mapping focus was either on large landslides (e.g. Caspian Sea KAZ, Owyhee USA, St. Elias USA) or on reconstructing historic landslide episodes from preserved geomorphic evidence (e.g. M8 Haiyuan CHN). Either way, the strategy for keeping such mapping practical is to use a size cutoff. Hence, although smaller landslides may be recognizable, they are excluded and thus censored in these inventories. Some of these catalogues used a size cutoff close to, or even above, our choice of $\mu = 0.1\ km^2$, such that undersampling of landslides near the size threshold may also explain some of the variance of the scaling estimates."*